# Nine-Year Surveillance of *Candida parapsilosis* Candidemia in a Cardiothoracic ICU: Insights into Mortality and Resistance

**DOI:** 10.3390/jof11100692

**Published:** 2025-09-23

**Authors:** Caio Trevelin Sambo, Bianca Leal de Almeida, Gabriel Fialkovitz, Tiago Alexandre Cocio, Afonso Rafael da Silva Junior, Lumena Pereira Machado Siqueira, Isabela Cristina Oliveira Silva, Flavia Rossi, Thaís Guimarães, Rinaldo Focaccia Siciliano, Evangelina da Motta Pacheco de Araújo, Gilda Maria Barbaro Del Negro, Gil Benard, Tania Mara Varejão Strabelli, Marcello Mihailenko Chaves Magri

**Affiliations:** 1Division of Infectious and Parasitic Diseases, Hospital das Clínicas, Faculdade de Medicina, Universidade de São Paulo, São Paulo 05403-010, Brazil; caio.sambo@hc.fm.usp.br (C.T.S.); isabelaoliveira87@gmail.com (I.C.O.S.); 2Hospital Infection Control and Infectious Diseases Service, Instituto do Cancer do Estado de Sao Paulo, Faculdade de Medicina, Universidade de São Paulo, São Paulo 01246-000, Brazil; bianca.almeida@hc.fm.usp.br; 3Infection Control Department, Instituto do Coração, Hospital das Clínicas, Universidade de São Paulo, São Paulo 05403-000, Brazil; gabriel.fialkovitz@hc.fm.usp.br (G.F.); rinaldo.siciliano@hc.fm.usp.br (R.F.S.); tania.strabelli@incor.usp.br (T.M.V.S.); 4Laboratory of Medical Mycology-LIM53, Instituto de Medicina Tropical, Divisão de Dermatologia, Hospital das Clínicas, Faculdade de Medicina, Universidade de São Paulo, São Paulo 05403-900, Brazil; alexcocio@gmail.com (T.A.C.); gildadelnegro@gmail.com (G.M.B.D.N.); bengil60@gmail.com (G.B.); 5Pathology Department, Laboratório de Microbiologia da Divisão de Laboratório Central, Hospital das Clínicas da Faculdade de Medicina USP (FMUSP), São Paulo 05403-010, Brazil; afonso.rafael@hc.fm.usp.br (A.R.d.S.J.); lumena.siqueira@hc.fm.usp.br (L.P.M.S.); f.rossi@hc.fm.usp.br (F.R.); evangelina.araujo@hc.fm.usp.br (E.d.M.P.d.A.); 6Infection Control Department, Hospital das Clínicas, University of São Paulo, São Paulo 05403-010, Brazil; tais.guimaraes@hc.fm.usp.br

**Keywords:** fluconazole-resistant *C. parapsilosis*, tertiary cardiothoracic hospital, candidemia

## Abstract

*Candida parapsilosis* has emerged as a prominent cause of nosocomial candidemia, particularly among critically ill patients. The increasing prevalence of fluconazole-resistant *C. parapsilosis* (FR-Cp) poses major therapeutic challenges, especially in resource-limited settings. We conducted a retrospective study of 144 patients with *C. parapsilosis* candidemia admitted to two post-surgical ICUs at a Brazilian tertiary cardiothoracic hospital between 2016 and August 2024. Demographic, clinical, microbiological, and therapeutic data were analyzed. Predictors of 30-day mortality were identified through multivariate logistic regression. The incidence density of *C. parapsilosis* candidemia ranged from 2.93 to 8.31 per 1000 hospitalizations. Fluconazole resistance was identified in 81% of isolates. Overall 30-day mortality was 55%. Independent risk factors for mortality included cardiopathy (OR: 19.36, *p* = 0.006), higher SOFA scores (OR: 1.54, *p* = 0.003), parenteral nutrition (OR: 29.77, *p* = 0.013), and dialysis (OR: 6.59, *p* = 0.043), while longer treatment duration was protective (OR: 0.81, *p* < 0.001). Fluconazole resistance was not independently associated with increased mortality. In this cohort of critically ill patients, *C. parapsilosis* candidemia was associated with high mortality and a high prevalence of fluconazole resistance. Clinical outcomes were mainly driven by host-related and therapeutic factors rather than antifungal resistance alone.

## 1. Introduction

Over the past decades hospital fungal infections have significantly increased, with *Candida* spp. accounting for approximately 80% of nosocomial fungal infections [1]. *C. albicans* has historically been the most prevalent species in nosocomial infections, with regional variations in prevalence rates [2,3]. However, non-*albicans Candida* species, particularly *C. parapsilosis*, have emerged as significant pathogens over recent decades, becoming a leading cause of candidemia worldwide [4]. In Brazil, studies at major hospitals, such as the Hospital das Clínicas of the University of São Paulo, have documented the evolving epidemiology of *C. parapsilosis*, highlighting its increasing incidence and its ability to form biofilms on medical devices, which is often linked to healthcare-associated transmission [5,6].

The emergence of fluconazole-resistant *C. parapsilosis* (FR-Cp) strains has become a major global concern. Although historically considered susceptible to azoles, increasing resistance has been reported in recent years in several regions, including Latin America, southern Europe, and Asia. High resistance rates have been described in single-center studies from Brazil (67.9%), South Africa (78%), Mexico (54%), Turkey (26.4%), Italy (33%), France (9.2%), and, more recently, Spain, where a sharp rise from 3.8% in 2019 to 29.1% in 2021 was documented. Similar trends have also been observed in other Mediterranean countries such as Greece. Resistance is mainly driven by *ERG11* gene substitutions, particularly Y132F, associated with clonal transmission and persistent hospital outbreaks [7,8,9,10]. A meta-analysis estimated a global fluconazole resistance prevalence of 15.2%, underscoring the emerging threat posed by FR-Cp [8].

Despite the growing recognition of *C. parapsilosis* as the second or third most prevalent cause of candidemia in Brazilian hospitals, data on its epidemiology, resistance patterns, and outcomes in critically ill cardiac patients remain limited. Specialized cardiologic ICUs face unique challenges, including high surgical complexity, frequent use of invasive devices, and prolonged hospital stays, which may contribute to increased fungal burden and antifungal resistance. This study retrospectively investigates candidemia caused by *C. parapsilosis* in two post-surgical ICUs at a major Brazilian tertiary cardiothoracic center.

## 2. Materials and Methods

This study aims to conduct a retrospective analysis of candidemia cases caused by *C. parapsilosis* in cardiothoracic patients in two post-surgical Intensive Care Units (ICU) at the Instituto do Coração do Hospital das Clínicas da Faculdade de Medicina da Universidade de São Paulo, a Brazilian tertiary care cardiothoracic hospital performing approximately 2000 cardiac surgeries annually. Ethical approval for the study was obtained from the local ethics committee (approval date: 3 July 2025; approval file: 7.686.849). The informed consent was waived because this study was retrospective, with the review of medical records.

Data collection was conducted through an active surveillance in a database maintained by the local hospital infection control service of the Heart Institute, which consolidates the microbiological results of infections recorded within the service. All patients with a positive blood culture for *C. parapsilosis* were included in the analysis. None of these patients were excluded on the final analysis. Candidemia was defined as the isolation of *C. parapsilosis* from blood culture, with only the first episode considered for analysis. A second episode was defined as a new positive blood culture occurring after a period of at least 30 days from the first episode [11]. Persistent candidemia was defined as positive blood cultures for *Candida* species lasting for ≥5 days [12]. Effective therapy was defined as the administration of a systemic antifungal agent with in vitro activity against *C. parapsilosis*, receiving at least 48 h of antifungal treatment. Optimal therapy was defined as the combination of an echinocandin and early catheter removal, the latter considered when the catheter was removed within ≤2 days after the candidemia diagnosis [13]. Prior antibiotic use was defined as the administration of broad-spectrum antibiotics, including carbapenems, ceftazidime-avibactam, polymyxins, aminoglycosides, glycopeptides, or linezolid. Prior antifungal use was defined as the administration of any azoles, echinocandins, or amphotericin. Both were considered in the last 30 days before candidemia.

This study comprehensively analyzed variables to characterize patients with candidemia caused by *C. parapsilosis* and to evaluate clinical outcomes in this population. Demographic data included patient age, reported as the mean ± standard deviation, and gender distribution, presented as the proportion of male patients. Clinical characteristics encompassed comorbidities such as cardiac disease, lung disease, diabetes mellitus (DM), chronic kidney disease, and immunosuppression, all documented if present at the time of hospital admission. Additionally, cases of concurrent COVID-19 infection during hospitalization were recorded. The clinical course was further characterized by complications associated with candidemia, including deep-seated infections, endophthalmitis, endocarditis, sternal osteomyelitis, and intra-abdominal infections. The study also evaluated the use of mechanical ventilation, vasoactive drug therapy, dialysis, and parenteral nutrition. Dialysis, total parenteral nutrition (TPN), and mechanical ventilation were considered if patients underwent these interventions up to the date of candidemia diagnosis. Sequential Organ Failure Assessment (SOFA) score and the use of vasoactive drugs were assessed on the date of candidemia diagnosis. Key laboratory findings included platelet count, creatinine levels, bilirubin levels, leukocyte count, neutrophil count, and C-reactive protein (CRP) levels within a 7-day window before or after the day of candidemia diagnosis. Echocardiography and fundoscopy were considered for analysis only if performed at least 72 h after the date of candidemia diagnosis. The study evaluated antifungal therapies prescribed, including fluconazole, echinocandins, and amphotericin B, as well as the adequacy of treatment.

The collected samples were sent to the Microbiology Laboratory of the Central Laboratory Division, Hospital das Clínicas, Faculdade de Medicina, University of São Paulo. Positive cultures were screened on chromogenic medium (ChromID *Candida*, bioMérieux, Marcy L’Étoile, France), and species identification was carried out by Matrix Assisted Laser Desorption Ionization Time-of-Flight mass spectrometry (MALDI-TOF MS) (Vitek MS, bioMérieux, Marcy-l’Étoile, France), and to FLC susceptibility testing according to Clinical and Laboratory Standards Institute guideline M44 [14]. Etest (bioMérieux, Marcy-l’Étoile, France) on RPMI 1640 agar was used to determine fluconazole MICs, and results were interpreted according to CLSI M27 (5th ed., 2022) and CLSI M60 (3rd ed., 2023) guidelines [15,16]. Microbiological data focused on the identification of *Candida* species in clinical cultures, with particular emphasis on *C. parapsilosis* and its resistance to fluconazole. The primary outcome of interest was the 30-day crude mortality rate, expressed as the percentage of deaths within 30 days of candidemia diagnosis.

A total of 17 *C. parapsilosis* isolates were selected for molecular analysis based on their availability and preservation in our laboratory strain collection, where representative isolates are routinely stored at −80 °C in glycerol stocks. Molecular analyses were performed at the Laboratory of Medical Mycology (LIM-53), Institute of Tropical Medicine/Division of Dermatology, Hospital das Clínicas, Faculdade de Medicina, University of São Paulo. *C. parapsilosis* sensu stricto (s.s.) isolates were genotyped by microsatellite analysis through amplification of seven different loci with primers previously described [17]. PCR products were separated on 3% agarose gel, stained with GelRed™ (Biotium, Fremont, CA, USA), and visualized with the UVITEC gel documentation system (Cleaver Scientific, Rugby, Warks, UK). For phylogenetic analysis, a .tif file generated by the photodocumentation system was used as input data for the PyEph 1.4 software (PyEph Project, 2025). Clustering was performed by the unweighted pair group method with arithmetic mean (UPGMA). The analysis also included the *C. parapsilosis* s.s. C935 belonging to the outbreak-related cluster described elsewhere [9]. In addition, the *C. parapsilosis* s.s. reference strain ATCC 22019 and two *C.parapsilosis* s.s. isolates (4E and 14D) unrelated to the outbreak cluster were also included to obtain the dendrogram.

The incidence density of candidemia (IDC) was calculated by dividing the number of candidemia cases recorded each year by the total number of hospitalizations in the same period, expressed as cases per 1000 hospitalizations. To test the association between categorical variables and hospital 30-day mortality, we used the χ2 test, Student’s *t*-test, or Fisher’s exact test when appropriate, describing results by frequency and percentage. Continuous variables are presented as mean and standard deviation and were compared using the Mann–Whitney U test. For multivariate analysis, we constructed a binary logistic regression model. Variables showing *p* < 0.1 in the univariate analysis were selected for inclusion; variables not meeting the prerequisites for logistic regression were excluded from the model. Values of *p* < 0.05 indicate statistical significance. Software used is IBM Corp. Released 2024. IBM SPSS Statistics for IOS, Version 30.0.0.0. Armonk, NY, USA: IBM Corp.

## 3. Results

This retrospective study included 144 cases of *C. parapsilosis* candidemia diagnosed in critically ill patients admitted to two cardiothoracic intensive care units of a Brazilian tertiary care hospital between 2016 and August 2024. The median age of the cohort was 51.1 years, with a predominance of male patients (54.1%). Fluconazole resistance was identified in 81% of *C. parapsilosis* isolates. The crude 30-day mortality rate was 55%.

The dendrogram analysis based on microsatellite profiles revealed four distinct clusters (Figure 1). One cluster consisting of all *C. parapsilosis* s.s. isolates evaluated in this study, indicating that they formed a single clonal lineage. The second cluster included the reference strain ATCC 22019, representing a distinct lineage with no epidemiological relationship to the studied isolates. In addition, a third and fourth clusters were formed by isolates 4E and 14D, characterizing lineages also unrelated to the studied isolates. These results demonstrate that the isolates were restricted to a single dominant *C. parapsilosis* s. s. lineage, genetically close to the C935 isolate, while the other clusters represented independent reference genotypes.

The IDC increased from 2.93 cases per 1000 hospitalizations in 2016 to a peak of 8.31 in 2020, followed by a gradual decline to 6.03 in 2021, 5.17 in 2022, and 5.52 in 2023 (Figure 2A). As of August 2024, the IDC was 5.21 cases per 1000 hospitalizations. *C. parapsilosis* accounted for the majority of candidemia episodes, comprising 144 out of 237 cases (60.8%). Annual frequencies of *C. parapsilosis* ranged from 9 cases in 2016 to a peak of 22 cases in 2021, declining to 8 cases in 2024 (up to August). The distribution of other *Candida* spp. during the study period was as follows: *C. albicans* accounted for 34 cases (14.3%), *C. tropicalis* for 32 cases (13.5%), *C. glabrata* for 11 cases (4.6%), *C. dubliniensis* for 6 cases (2.5%), *C. guilliermondii* for 3 cases (1.3%), *C. krusei* for 4 cases (1.7%), *C. haemulonii* for 2 cases (0.8%), and *C. lusitaniae* for 1 case (0.5%). The temporal trends of all *Candida* spp. are summarized in Figure 2B.

The univariate analysis revealed that patients who died were older (63.1 ± 16.8 years) compared to survivors (58.1 ± 19.3 years), (*p* = 0.016, OR: 1.02 [99% CI: 1.00–1.03]). Gender did not show a significant association with mortality, as 70% of deceased patients were male, compared to 54.1% overall (*p* = 0.575, OR: 0.91 [99% CI: 0.68–1.23]). The length of stay in the ICU was notably shorter among patients who died (mean 52 days) compared to those who survived (mean 91 days), a highly significant finding (*p* < 0.001). Patient demographics, comorbidities, and clinical features by 30-day outcome are shown in Table 1.

Among comorbidities, cardiac disease was significantly associated with 30-day mortality (*p* = 0.003, OR: 2.30 [99% CI: 1.05–5.01]). Other comorbidities, including immunosuppression (*p* = 0.928, OR: 1.01 [99% CI: 0.71–1.45]), DM (*p* = 0.517, OR: 0.88 [99% CI: 0.65–1.21]), chronic kidney disease (*p* = 0.226, OR: 1.87 [99% CI: 0.67–5.25]), lung disease (*p* = 0.817, OR: 0.88 [99% CI: 0.32–2.80]) and COVID-19 (*p* = 0.207, OR: 0.89 [99% CI: 0.56–1.41]), did not show significant associations with mortality.

Endocarditis had an OR of 0.46 [99% CI: 0.10–2.00], not statistically significant (*p* = 0.290). Endophthalmitis occurred in only one patient who died, limiting statistical analysis.

Patients requiring vasoactive drugs had significantly higher odds of death (*p* < 0.001, OR: 4.57 [99% CI: 2.04–10.25]), as did those on mechanical ventilation (*p* = 0.03, OR: 2.82 [99% CI: 1.42–5.60]). Dialysis was more common in deceased patients (*p* = 0.019, OR: 2.22 [99% CI: 1.13–4.37]), and parenteral nutrition showed a strong association with mortality (*p* = 0.007, OR: 3.93 [99% CI: 1.38–11.71]).

There was no statistically significant association between fluconazole resistance and 30-day mortality (*p* = 0.113, OR: 1.07 [99% CI: 0.71–1.62]). The proportion of resistant isolates was similar among survivors (54%) and deceased patients (57%), suggesting that fluconazole resistance alone may not be a primary determinant of mortality in this cohort. Figure 2C shows the annual distribution of fluconazole-resistant and susceptible *C. parapsilosis* isolates throughout the study period. Prior exposures, microbiological findings, treatment variables, and laboratory results by outcome are summarized in Table 2.

Effective antifungal therapy was a protective factor, with survivors receiving it in 94% of cases compared to 82% among deceased patients (*p* = 0.043, OR: 1.53 [99% CI: 1.12–2.08]). Timely catheter removal was also significantly protective (*p* < 0.001, OR: 2.06 [99% CI: 1.50–2.83]), although early removal showed no significant association (*p* = 0.700, OR: 0.89 [99% CI: 0.51–1.55]). The choice of antifungal treatment (fluconazole, echinocandins, or amphotericin B) did not show significant associations with mortality (*p* > 0.05).

Platelet counts were significantly lower in patients who died (119,375 ± 99,317) compared to survivors (210,265 ± 119,541; *p* < 0.001). Bilirubin levels were markedly higher in deceased patients (3.75 ± 5.24 mg/dL) than in survivors (0.77 ± 0.54 mg/dL; *p* < 0.001).

The time between candidemia diagnosis and death was significantly shorter for deceased patients (10.9 ± 8.14 days) compared to survivors (56.7 ± 32.7 days; *p* < 0.001). Conversely, total treatment duration was significantly longer for survivors (33 ± 25.4 days) compared to deceased patients (13.3 ± 9.7 days; *p* < 0.001). The time between surgery and candidemia was shorter among deceased patients (*p* = 0.013), and the time to first negative blood culture was significantly shorter for deceased patients (*p* = 0.004).

Multivariate logistic regression identified five independent factors associated with 30-day mortality (Table 3). Pre-existing cardiopathy (OR = 19.356, 95% CI: 2.293–163.3, *p* = 0.006), higher SOFA scores (OR = 1.54, 95% CI: 1.158–2.049, *p* = 0.003), parenteral nutrition (OR = 29.767, 95% CI: 2.032–436.0, *p* = 0.013), and dialysis (OR = 6.59, 95% CI: 1.059–41.162, *p* = 0.043) were associated with increased mortality risk. Conversely, longer treatment duration was protective (OR = 0.807, 95% CI: 0.717–0.907, *p* < 0.001). These findings highlight the interplay of clinical factors influencing outcomes in ICU patients with candidemia.

## 4. Discussion

The results of this study provide valuable insights into the epidemiology and clinical outcomes of candidemia caused by *C. parapsilosis* in cardiologic critically ill patients, particularly in a resource-limited setting in Brazil. The study revealed a marked increase in incidence density during the COVID-19 pandemic, peaking in 2020, followed by a subsequent decline. Furthermore, the predominance of *C. parapsilosis* as the main pathogen and the alarming fluconazole resistance rate of 81% observed in this study highlight the unique challenges faced in managing candidemia in this region.

The incidence of candidemia varies significantly across different geographic regions, healthcare settings, and patient populations. In this study, the incidence density of candidemia caused by *C. parapsilosis* ranged from 2.93 cases per 1000 hospitalizations in 2016 to a peak of 8.31 cases per 1000 hospitalizations in 2020, with a subsequent decline in the following years. These findings contrast with a study from the same center [6], which reported incidence rates of up to 2.3 per 1000 hospital admissions and up to 1.3 per 1000 admissions in a tertiary care hospital in Rio de Janeiro [18]. Comparatively, data from developed countries show similar incidence rates when overall candidemia is compared to *C. parapsilosis* candidemia in this study. For instance, a multicenter study in Europe reported a cumulative incidence of 7.07 episodes per 1000 ICU admissions, with candidemia accounting for 5.52 episodes per 1000 admissions [19]. The predominance of *C. parapsilosis* in this study differs with trends observed in Brazil and other middle-income countries. While in this study *C. parapsilosis* accounted for 60% of candidemia cases in other studies *C. albicans* remains the main species accounting for 44%, 39% and 41.7% [6,20,21].

The temporal trends observed in this study, including the peak incidence during 2020, coincide with the impact of the COVID-19 pandemic. Increased ICU admissions, prolonged mechanical ventilation, and widespread use of immunosuppressive therapies likely contributed to the observed rise in cases [22,23]. This aligns with broader trends in Brazil, notably a 38% rise in candidemia incidence density reported by Jordana et al. [6] during 2020 compared to pre-pandemic years (2016–2019), reinforcing the pandemic’s role in exacerbating fungal threats.

Regarding *C. parapsilosis* resistance, this study found 81% fluconazole resistant strains, a high burden even when compared to other reports in Brazil, where fluconazole resistance in *C. parapsilosis* has been documented to range from 6–11% and up to 26% in a study performed in Brazil’s Midwest region particularly in tertiary care hospitals [24]. In 2022, Thomaz et al. described an interhospital transmission event between the Oncology Center of Hospital das Clínicas and the Cardiology Center of the present study during the COVID-19 pandemic in 2020 [23]. At that time, fluconazole-resistant strains accounted for 60% of isolates in the Cardiology Center, a profile that persisted throughout the years covered by this study, which reported an even higher resistance rate of 81%.

Despite the high prevalence of resistance, this study found no significant association between fluconazole resistance and 30-day mortality. In other study that evaluated FR-Cp during de COVID-19 pandemic showed 46.7% vs. 57.5% 30-day crude mortality between fluconazole resistant and susceptible groups with no statistical difference (*p* = 0.312) [22]. Also, in a French retrospective analysis of 78 invasive infections by *C. parapsilosis*, 30-day all-cause mortality between fluconazole susceptible and resistant strains were, respectively, 22.4% and 45.5% but also with no statistical significance (*p* = 0.082) [25]. An outbreak report of candidemia by fluconazole-resistant strains showed statistically significant difference regarding all-cause mortality (63.8% x 20%, *p* = 0.008) [26]. The lack of association in some studies may be attributed to the early use of alternative antifungals or to the predominance of fluconazole-resistant strains with lower virulence compared to other *Candida* species. Further studies are needed to evaluate the clinical and economic impact of FR-Cp candidemia.

In the present study, the overall 30-day mortality rate for candidemia was 55%, highlighting the significant burden of this infection in critically ill patients. This finding aligns with previous Brazilian studies, which report mortality rates up to 63.8% in tertiary centers [9]. A study in Turkey showed 50% mortality in candidemia by fluconazole-resistant strains [27]. A lower mortality rate from *C. parapsilosis* infection was noted in a prospective, population-based study among adults hospitalized in medical and surgical ICUs throughout Spain. The rate of death within 7 days for *C. parapsilosis* infection was 7%, versus 56% for *C. albicans,* which differs significantly from the present study [28].

Previous studies in Brazil that emphasize the role of organ dysfunction scores in predicting outcomes in overall candidemia. Araújo et al., 2024 [6] showed the following risk factors for 30-day mortality: advanced age, lung disease, heart disease, ICU stay, hemodialysis, use of corticosteroids, use of antibiotics, and mechanical ventilation. Colombo et al. 2012 showed association with higher 30-day mortality by univariate analysis: older age, period 1, higher APACHE II score, cancer, lung disease, renal failure, dialysis, mechanical ventilation, receipt of corticosteroids, no treatment for candidemia and treatment with deoxycholate amphotericin B. Infection due to *C. parapsilosis* though in that study was associated with lower mortality [21]. The only comorbidity associated with higher 30-day mortality was cardiac disease in both univariate and multivariate analysis. And this is also shown in other studies [6,21]. The multivariate analysis further refined the understanding of mortality predictors by identifying independent factors significantly associated with 30-day mortality. Among these, cardiac disease emerged as a key risk factor (*p* = 0.006, OR = 19.356), underscoring the vulnerability of patients with cardiovascular comorbidities. SOFA score remained a significant predictor in the multivariate model (*p* = 0.003, OR = 1.54), confirming the importance of systemic organ dysfunction as a determinant of outcomes. Other retrospective studies also highlighted how increased severity scores are associated with mortality, one which showed an increased risk when APACHE II score higher than 25 (OR 43.9) [28].

The need for parenteral nutrition and dialysis were also independently associated with mortality in the multivariate analysis, reinforcing their roles as indicators of critical illness. Parenteral nutrition has been identified as an independent risk factor for candidemia, including cases caused by *Candida parapsilosis*. The study by Poissy et al. demonstrated that total parenteral nutrition is an independent risk factor for candidemia in both ICU patients and those admitted to other hospital units [25]. Additionally, the study by Kutlu et al. also found a significant association between the use of parenteral nutrition and mortality in patients with candidemia [27].

The duration of antifungal treatment demonstrated a significant protective effect in our study (*p* < 0.001, OR = 0.807), highlighting the importance of prompt and adequate therapy. This finding aligns with previous research advocating for early and sustained antifungal treatment to improve outcomes in candidemia [21]. In our study, the failure to remove the catheter in the univariate analysis was associated with a higher risk of death (*p* < 0.001). Several studies have emphasized the critical role of early antifungal therapy and catheter removal in candidemia management. For instance, early appropriate antifungal treatment was shown to reduce mortality to 34.9%, and the combination of antifungal therapy with catheter removal within 48 h further decreased mortality to 18.9% (HR 0.34; 95% CI 0.16–0.70; *p* = 0.03) [28]. Similarly, a Spanish cohort study demonstrated that early antifungal therapy and catheter removal were independently associated with reduced early mortality (OR 0.27; 95% CI 0.08–0.91) [29]. These findings, consistent with our results, underscore the importance of timely interventions, particularly early antifungal therapy and catheter management, as essential components of candidemia treatment strategies to improve patient outcomes.

This study has several limitations that should be acknowledged. First, its retrospective nature may introduce biases related to data collection and the completeness of medical records. Missing data on certain variables, such as specific antifungal dosing, timing of catheter removal, and additional comorbidities, could have impacted the analysis of some risk factors and treatment outcomes. Second, the study was conducted in a single tertiary care hospital, which may limit the generalizability of the findings to other settings. Differences in healthcare infrastructure, patient demographics, and antifungal stewardship practices in other institutions and regions might result in variations in the epidemiology and outcomes of candidemia. Third, the high prevalence of fluconazole resistance observed in this study reflects the local epidemiological context and antifungal usage patterns, which might not be representative of other regions or countries. This limits the ability to extrapolate these findings to settings with lower rates of resistance or differing *Candida* species distributions. Another limitation is the relatively small sample size for certain subgroup analyses, such as the association between fluconazole resistance and mortality. Larger multicenter studies would provide a more robust assessment of the clinical impact of resistance and other factors. Another limitation of this study is that molecular analyses to detect mutations associated with azole resistance were not performed, as they are not part of our laboratory’s routine workflow. Antifungal susceptibility testing was performed only for fluconazole, and other antifungal agents were not tested. Despite these limitations, the study provides valuable insights into the epidemiology, resistance patterns, and outcomes of candidemia in critically ill patients, particularly in the Brazilian healthcare context. These findings contribute to a better understanding of this challenging infection and highlight areas for improvement in clinical management and infection control.

## 5. Conclusions

This study highlights the significant burden of *C. parapsilosis* candidemia in critically ill patients, showing high rates of fluconazole resistance and mortality. While the high prevalence of fluconazole-resistant isolates is concerning, it was not independently associated with increased mortality, suggesting that early and appropriate antifungal treatment may mitigate its impact. The findings underscore the importance of continued antifungal stewardship, targeted infection control strategies, and early therapeutic interventions to improve outcomes in patients with candidemia. Furthermore, the study highlights the need for regional surveillance and multicenter collaborations to better understand the evolving epidemiology of *C. parapsilosis* candidemia, particularly in resource-limited settings like Brazil. Future prospective studies should aim to address the limitations identified, explore interventions to improve patient outcomes, and further investigate the clinical and microbiological factors contributing to mortality.

## Figures and Tables

**Figure 1 jof-11-00692-f001:**
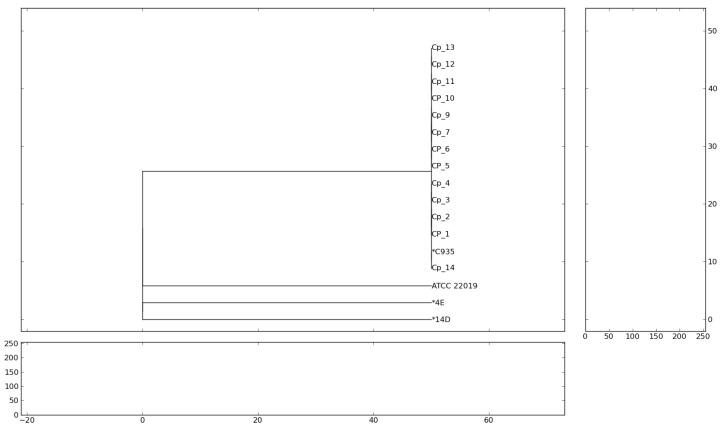
Dendrogram of *C. parapsilosis* s.s. isolates based on microsatellite profiles using the UPGMA method. Dendrogram showing the genetic relationship among the *C. parapsilosis* s.s. isolates, * 935C, 4E, 14D and the reference strain ATCC 22019, constructed using concatenated allele profiles from seven microsatellite loci and clustered by the UPGMA method.

**Figure 2 jof-11-00692-f002:**
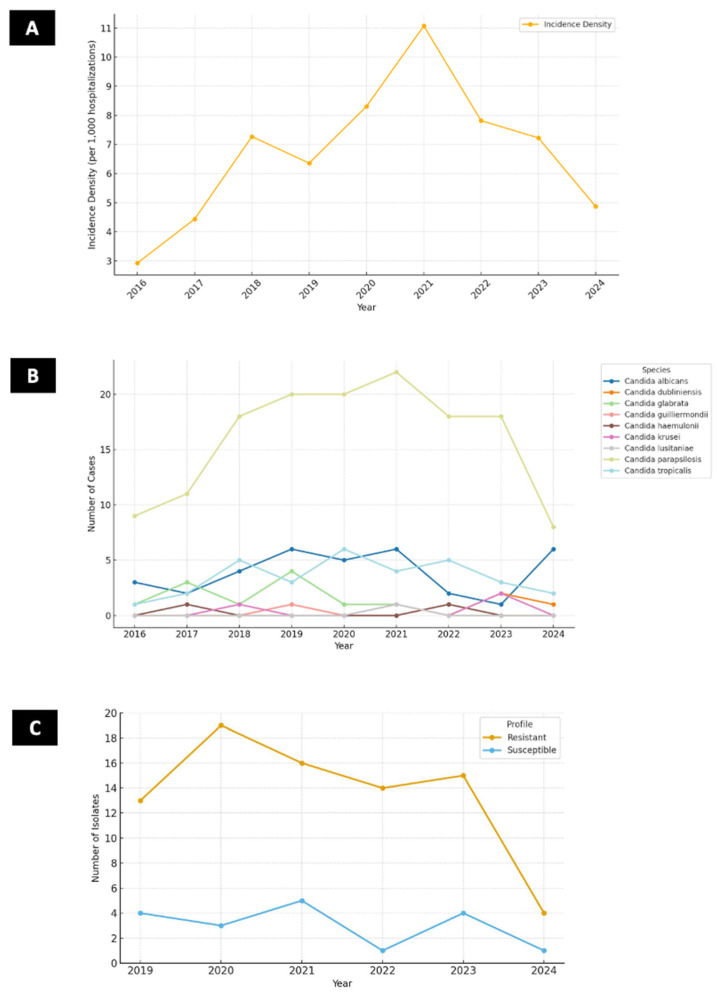
Temporal trends of candidemia episodes and fluconazole resistance in a cardiothoracic ICU (2016–2024). (**A**) Incidence density of candidemia episodes per 1000 hospitalizations over the study period. (**B**) Annual distribution of Candida species causing bloodstream infections. (**C**) Annual number of *C. parapsilosis* bloodstream isolates according to fluconazole susceptibility profile (resistant vs. susceptible).

**Table 1 jof-11-00692-t001:** Univariate Analysis of Demographic, Comorbidity, and Clinical Variables Associated with 30-Day Mortality in ICU Patients with *Candida parapsilosis* Candidemia.

Variable	All Patients (144)	Alive (64)	Dead (80)	*p*	OR (IC 99%)
Age (mean, ST; years)	51.1 ± 22.6	58.1 ± 19.3	63.1 ± 16.8	0.360	1.02 (1.003–1.03)
Male	78 (54.1%)	33 (30%)	45 (70%)	0.575	0.91 (0.68–1.23)
Intensive Care Unit Admission	144 (100%)	64 (100%)	80 (100%)	-	-
Length of Stay	89 ± 62.8	91	52	<0.01	-
Comorbidities					
Cardiac disease	109 (75%)	43 (67%)	66 (82%)	0.003	2.30 (1.05–5.01)
Imunossupression	31 (21%)	14 (21%)	17 (21%)	0.928	0.96 (0.43–2.14)
Lung disease	17 (11%)	8 (12%)	9 (11%)	0.817	0.88 (0.32–2.44)
Diabetes mellitus	38 (26%)	15 (23%)	23 (28%)	0.472	1.31 (0.62–2.80)
Chronic Kidney disease	19 (13%)	6 (9%)	13 (16%)	0.226	1.87 (0.67–5.25)
COVID-19	13 (9%)	5 (7%)	8 (10%)	0.649	1.31 (0.40–4.22)
Surgery	132 (91%)	59 (92%)	73 (91%)	0.840	0.88 (0.26–2.92)
Complications					
Deep Site Infection (n=141)	31 (22%)	15 (23%)	16 (20%)	0.638	1.09 (0.74–1.59)
Endophthalmitis (n = 24)	1 (4%)	0	1 (4.3)	-	-
Endocarditis	8 (5%)	5 (7%)	3 (3%)	0.290	0.46 (0.10–2.00)
Sternal osteomyelitis	15 (10%)	6 (9%)	9 (11%)	0.714	1.22 (0.41–3.64)
Intra-abdominal infection	2 (1%)	0	2 (2.5%)	0.203	0.54 (0.47–0.63)
Surgical Site infection	18 (12%)	10 (15%)	8 (10%)	0.310	0.60 (0.22–1.62)
Persistent Candidemia (n = 80)	26 (32%)	14 (21%)	12 (15%)	0.536	0.84 (0.49–1.43)
Clinical characteristics					
Fever	29 (20%)	15 (23%)	14 (17.5%)	0.68	0.82 (0.28–2.41)
Source control (n = 36)	36 (25%)	18 (50%)	18 (50%)	0.14	2.12 (1.48–3.03)
Dialysis	83 (57%)	30 (46%)	53 (66%)	0.019	2.22 (1.13–4.37)
Vasoactive drug	106 (73%)	37 (57%)	69 (86%)	<0.01	4.57 (2.04–10.25)
SOFA (n = 110)	7.6 ± 4.2	4.83 ± 3.3	9.59 ± 3.72	<0.01	-
Mean arterial pression (n = 128)	70.15 ± 18.7	70.4 ± 20.7	69.9 ±17.2	0.54	-
Mechanical Ventilation	83 (57%)	28 (43%)	55 (68%)	0.003	2.82 (1.42–5.60)
Bacteremia	38 (26%)	18 (28%)	20 (25%)	0.179	1.07 (0.76–1.51)
Parenteral Nutrition	25 (17%)	5 (7%)	20 (25%)	0.007	3.93 (1.38–11.17)

SOFA = Sequential Organ Failure Assessment.

**Table 2 jof-11-00692-t002:** Univariate Analysis of Treatment, Microbiological, and Laboratory Factors Associated with 30-Day Mortality in ICU Patients with *Candida parapsilosis* Candidemia.

Variable	All Patients (144)	Alive (64)	Dead (80)	*p*	OR (IC 99%)
Prior drug exposure					
Prior Antibiotics	140 (97%)	62 (96%)	78 (97.5%)	0.821	1.25 (0.17–9.18)
Prior antifungal	71 (49%)	27 (42%)	44 (55%)	0.126	1.67 (0.86–3.25)
Microbiology					
*C.parapsilosis* Fluconazole-Resistant (n = 99)	81 (81%)	35 (54%)	46 (57%)	0.113	1.07 (0.71–1.62)
>1 Candida on the same blood culture	3 (2%)	1 (1,5%)	2 (2.5%)	0.684	1.35 (0.31–5.90)
>1 Candida during CP candiemia	8 (5%)	3 (4%)	5 (6%)	0.684	0.88 (0.19–7.45)
*C.albicans*	5 (3%)	2 (3%)	3 (3%)	0.839	1.20 (0.44–1.91)
*C.tropicalis*	5 (3%)	1 (1.5%)	4 (5%)	0.263	3.31 (0.36–30.42)
*C.glabrata*	1 (0.6%)	0	1 (1.25%)	0.369	0.55 (0.47–0.64)
Prior *C.parapsilosis* colonization	42 (29%)	17 (26%)	25 (31%)	0.378	0.90 (0.66–1.23)
Treatment					
Antifungal prescribed	124 (86%)	55 (85%)	69 (86%)	1.000	1.03 (0.20 -3.58)
Effective therapy	109 (75%)	52 (94%)	57 (82%)	0.043	1.53 (1.12–2.08)
Fluconazole (n = 124)	28 (22%)	13 (23%)	15 (21%)	0.063	1.05 (0.71–1.54)
Echinocandin (n = 124)	91 (73%)	39 (71%)	52 (75%)	0.311	0.90 (0.61–1.31)
Amphotericin (n = 124)	8 (6%)	3 (5.5%)	5 (7.2%)	0.163	0.83 (0.50 –1.54)
Optimal Therapy	33 (22%)	22 (14%)	11 (18%)	0.003	1.53 (1.12–2.08)
Catheter removal (n = 140)	77 (55%)	48 (62%)	29 (37%)	<0.01	2.06 (1.50–2.83)
Early catheter removal (n = 81)	37 (45%)	22 (59%)	14 (40%)	0.700	0.89 (0.51–1.55)
Laboratory Findings					
Platelets (×10^3^/mm^3^)	159.770 ± 117.469	210.265 ± 119.541	119.375 ± 99.317	<0.01	-
Creatinine (mg/dL)	1.85 ± 1.24	1.68 ± 1.24	1.98 ± 1.24	0.038	-
Bilirrubins (n = 111)	2.54 ± 4.3	0.77 ± 0.54	3.75 ± 5.24	<0.01	-
Leucocytes	13.571 ± 8319	13.10 ± 7035	13.948 ± 9246	0.887	-
Neutrophils	10.668 ± 7095	10.474 ± 6475	10.823 ± 7593	0.963	-
PCR [(mg/L (n = 142)]	127 ± 75.8	103.6 ± 80.69	147.45 ±93.98	0.156	-
Time frames					
Days between candidemia and death	31.2 ± 32.1	56.7 ± 32.7	10.9 ± 8.14	<0.01	-
Days between hospitalization and candidemia	57.8 ± 49.3	55.08 ± 49.7	59.9 ± 49.2	0.604	-
Days between surgery and candidemia (n = 133)	28.7 ± 102.3	30.6 ± 49.2	27.2 ± 130.4	0.013	-
Days between candidemia and treatment (n = 86)	11.69 ± 78.56	2.32 ± 1.77	14.34 ± 88.97	0.371	-
Days between candidemia and first negative blood culture (n = 72)	18.7 ± 62	25.6 ± 74.4	4.17 ± 2.61	0.004	-
Days between candidemia and catheter removal (n = 79)	5.4 ± 34	1.5 ± 13.6	11.7 ± 52.9	0.406	-
Total treatment duration (n = 123)	22 ± 20.7	33 ± 25.4	13.3 ± 9.7	<0.01	-
30-Day crude mortality	80 (55%)	-	-	-	-

SOFA = Sequential Organ Failure Assessment. Colonization was defined as the presence of *Candida* species in cultures from non-sterile sites without signs or symptoms of active infection. PCR = Reactive C protein.

**Table 3 jof-11-00692-t003:** Multivariate analysis for mortality in 30 days.

Variable	*p*	OR	CI 95%
Cardiac disease	0.006	19.356	2.293–163.3
SOFA	0.003	1.54	1.158–2.049
Parenteral nutrition	0.013	29.767	2.032–436.0
Dialysis	0.043	6.59	1.059–41.162
Total treatment duration	<0.001	0.807	0.717–0.907

SOFA = Sequential Organ Failure Assessment.

## Data Availability

Anonymized data may only be available upon well detailed and pertinent request due to privacy or ethical restrictions. Please contact the corresponding authors.

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
