# Peer review of "Nine-Year Surveillance of *Candida parapsilosis* Candidemia in a Cardiothoracic ICU: Insights into Mortality and Resistance"

_jof, 2025, doi:10.3390/jof11100692_

Round 1

Reviewer 1 Report

In this manuscript, the Authors perform a retrospective study of cases of Candida parapsilosis in their Cardiothoracic ICU. Over 9 years, there were 144 cases of C. parapsilosis candidemia, with a high rate of fluconazole resistant isolates (81%) and a 30-day mortality rate of 55%. Host-related factors were associated with increased mortality, not resistance to fluconazole.

Overall I found this manuscript well-written and insightful.

  1. Table 1 is very large. I would consider breaking this down into multiple tables.
  1. Page 3, line 107: Please define what SOFA means for the readers.
  2. Page 4, lines 159-162: Please rephrase this sentence. What do you mean by "These results demonstrate that the was restricted..."
  3. Page 4, lines 165-166: C. parapsilosis accounted for most cases of what? Candidemia? Please rephrase. 
  4. Page 5, lines 198-200: What do the Authors consider the definition of "effective" antifungal therapy? Was this based on MIC data?
  5. Page 5, lines 220-222: Please remove this sentence
  6. Page 12, line 280: Do you mean "and 1.3 per 1,000 admissions in tertiary care hospitals."?
  7. Page 11 and 12: Lines 268-271 and 289-292 very closely say the same thing. I would consider removing one of these sentences

Author Response

RESPONSE TO THE REVIEWERS

Manuscript ID: jof-3874742

Title: “Nine-Year Surveillance of Candida parapsilosis Candidemia in a Cardiothoracic ICU: Insights into Mortality and Resistance”

REVIEWER 1

In this manuscript, the Authors perform a retrospective study of cases of Candida parapsilosis in their Cardiothoracic ICU. Over 9 years, there were 144 cases of C. parapsilosis candidemia, with a high rate of fluconazole resistant isolates (81%) and a 30-day mortality rate of 55%. Host-related factors were associated with increased mortality, not resistance to fluconazole.

Overall I found this manuscript well-written and insightful.

  1. Table 1 is very large. I would consider breaking this down into multiple tables.

Answer:  We sincerely thank the reviewer for this valuable observation and for the overall positive evaluation of our manuscript. We agree that Table 1 was extensive and could be difficult to navigate. To improve readability and clarity, we have reorganized the table by grouping related variables into thematic sub-sections (demographic characteristics, comorbidities, clinical features, microbiological findings, treatment variables, and outcomes). These changes have been implemented in the revised version of the manuscript (now Table 1, 2).

  1. Page 3, line 107: Please define what SOFA means for the readers.

Answer: We thank the reviewer for this helpful suggestion. We have now defined the acronym at first mention in the text as follows: “SOFA (Sequential Organ Failure Assessment) score.”

  1. Page 4, lines 159-162: Please rephrase this sentence. What do you mean by "These results demonstrate that the was restricted..."

Answer: We thank the reviewer for pointing out this unclear sentence. We have revised it for clarity as follows: “These results demonstrate that the isolates were restricted to a single dominant C. parapsilosis sensu stricto lineage, genetically close to the C935 isolate, while the other clusters represented independent reference genotypes.” This revision has been incorporated into the Results section (page 4, lines 157–160) of the revised manuscript.

  1. Page 4, lines 165-166: C. parapsilosis accounted for most cases of what? Candidemia? Please rephrase. 

Answer: We thank the reviewer for this clarification. The sentence has been revised for accuracy and specificity as follows: “C. parapsilosis accounted for the majority of candidemia episodes, comprising 144 out of 237 cases (60.8%).”

  1. Page 5, lines 198-200: What do the Authors consider the definition of "effective" antifungal therapy? Was this based on MIC data?

Answer: We thank the reviewer for this important question. In our study, effective antifungal therapy was defined as the administration of an antifungal agent with in vitro activity against C. parapsilosis for at least 48 hours, initiated after the candidemia diagnosis.

  1. Page 5, lines 220-222: Please remove this sentence

Answer: We thank the reviewer for this suggestion. The requested sentence has been removed from the Results section (page 5, lines 220–222) in the revised manuscript.

  1. Page 12, line 280: Do you mean "and 1.3 per 1,000 admissions in tertiary care hospitals."?

Answer: We thank the reviewer for this observation. Yes, that is correct. We have revised the sentence for clarity as follows: “These findings contrast with a study from the same center [6], which reported incidence rates of up to 2.3 per 1,000 hospital admissions and up to 1.3 per 1,000 admissions in a tertiary care hospital in Rio de Janeiro [15].”

  1. Page 11 and 12: Lines 268-271 and 289-292 very closely say the same thing. I would consider removing one of these sentences.

Answer: We thank the reviewer for this valuable observation. We agree that the two sentences were redundant. To improve clarity and conciseness, we have removed the sentence in lines 268–271 and retained the one in lines 289–292, which provides a clearer link to the COVID-19 pandemic. This change has been incorporated into the revised Discussion section (pages 11–12).

Reviewer 2 Report

Review

Nine-Year Surveillance of Candida parapsilosis Candidemia in a Cardiothoracic ICU: Insights into Mortality and Resistance

The study by Trevelin Sambo and colleagues report a very interesting epidemiological retrospective survey on Candidemia episodes in a Cardiothoracic ICU, with a specific focus on Candida parapsilosis. This article is a knowledgeable work, with important findings. However, authors must assess the issues reported below

Detailed comments:

Abstract must be unstructured and a maximum of 200 words, please make corrections accordingly.

Introduction.

Line 53-58. The epidemiologic background needs to be expanded. The countries reported are not the only ones facing clinical outbreaks of Fluconazole-Resistant C. parapsilosis, most of the mediterranean countries as Greece, Italy, Turkey and Spain are also struggling with this nosocomial fungal pathogen, with worrisome rates of fluconazole resistance and new mutations discovered in such clinical scenarios. Please expand this section adding the proper references.

Line 55. Genes must be reported in italics

Methods.

Line 85-86. This sentence is confusing. What in the case of administration of an antifungal to which the isolate proved to be resistant at antifungal susceptibility testing? Do the authors have a reference to justify such definition of “Effective therapy”?

Line 115-123. No description of the antifungal susceptibility testing used was reported, please specify the test and the reference for interpreting MIC results. This is a major issue. How could the authors state that isolates were resistant to Fluconazole if there is no reference and no description of the testing method? Plus, did the authors assess mutations known to be associated with fluconazole resistance in Candida parapsilosis as the Y132F in the ERG11, the K143R, R398I, G650E in TAC1, G583R and S862C in the CpMRR1 etc. If lab routine workflow does include searching for these mutations associated with azole-resistance it should be acknowledged.

Line 115-123. Were any genome sequencing analyses performed? If not please acknowledge.

Results.

Line 152. What about the other antifungal molecules tested? Were voriconazole and posaconazole affected too?

Authors should report antifungal susceptibility profiles of all molecules tested.

Did fluconazole resistance increase over-time? Authors should depict antifungal resistance incidence rate over time in a separate figure, this is a very important aspect that in a study like this has to be fully elucidated.

Figure 1. How many isolates were included in the dendrogram study? How did the authors select these isolates? If this is a retrospective study how was it possible to select such isolates? Authors must explain this important part of the study in both results and methods section.

Figure 3. Candida parapsilosis and Candida albicans are indistinguishable, please correct the figure accordingly.

Table 1. this table is too big, either authors place it in a supplementary table or divide it into separate tables.

Discussion.

Line 272. Is 81% a prevalence or a resistance rate? From my understanding this percentage is an overall calculation, however incidence rates have not been reported or assessed through the study. Authors are encouraged to better clarify resistance rates and trends overtime as well as clarifying and describing resistance rate for other molecules tested.

Author Response

REVIEWER 2

The study by Trevelin Sambo and colleagues report a very interesting epidemiological retrospective survey on Candidemia episodes in a Cardiothoracic ICU, with a specific focus on Candida parapsilosis. This article is a knowledgeable work, with important findings. However, authors must assess the issues reported below

Detailed comments:

  • Abstract must be unstructured and a maximum of 200 words, please make corrections accordingly.

Answer: We thank the reviewer for this important observation. We have revised the Abstract to comply with the journal’s formatting requirements, presenting it as an unstructured paragraph and reducing its length to 197 words. The updated Abstract has been included in the revised version of the manuscript.

Introduction

  • Line 53-58. The epidemiologic background needs to be expanded. The countries reported are not the only ones facing clinical outbreaks of Fluconazole-Resistant C. parapsilosis, most of the mediterranean countries as Greece, Italy, Turkey and Spain are also struggling with this nosocomial fungal pathogen, with worrisome rates of fluconazole resistance and new mutations discovered in such clinical scenarios. Please expand this section adding the proper references.

Answer: We sincerely thank the reviewer for this valuable comment. We have expanded the epidemiological background to include other affected countries, particularly from the Mediterranean region, where C. parapsilosisoutbreaks associated with high fluconazole resistance rates and ERG11 mutations (mainly Y132F) have been reported. This additional context has been incorporated into the Introduction section (lines 51–60) of the revised manuscript.

  • Line 55. Genes must be reported in italics

Answer: We thank the reviewer for this important observation. We have revised the manuscript to ensure that all gene names (e.g., ERG11, TAC1, MRR1) are now consistently presented in italics throughout the text, in accordance with journal guidelines

Methods

  • Line 85-86. This sentence is confusing. What in the case of administration of an antifungal to which the isolate proved to be resistant at antifungal susceptibility testing? Do the authors have a reference to justify such definition of “Effective therapy”?

Answer: We thank the reviewer for this important comment and the opportunity to clarify this point. Effective therapy was defined as the administration of a systemic antifungal agent with in vitro activity against C. parapsilosis, receiving at least 48 hours of antifungal treatment. The revised text now reads as follows:

“Optimal therapy was defined as the combination of an echinocandin and early catheter removal, the latter considered when the catheter was removed within ≤2 days after the candidemia diagnosis [13].”This change has been incorporated into the Methods section

  • Line 115-123. No description of the antifungal susceptibility testing used was reported, please specify the test and the reference for interpreting MIC results. This is a major issue. How could the authors state that isolates were resistant to Fluconazole if there is no reference and no description of the testing method?

Answer: We thank the reviewer for this important comment. We have now added a detailed description of the antifungal susceptibility testing method used, as well as the reference guidelines for interpreting fluconazole MIC results. The revised text in the Methods section now reads as follows:

“...and to FLC susceptibility testing according to Clinical and Laboratory Standards Institute guideline M44 [31]. Etest (bioMérieux, Marcy-l’Étoile, France) on RPMI 1640 agar was used to determine fluconazole MICs, and results were interpreted according to CLSI M27 (5th ed., 2022) and CLSI M60 (3rd ed., 2023) guidelines [32,33]. Microbiological data focused on the identification of Candida species in clinical cultures, with particular emphasis on C. parapsilosis and its resistance to fluconazole.”

  • Did the authors assess mutations known to be associated with fluconazole resistance in Candida parapsilosis as the Y132F in the ERG11, the K143R, R398I, G650E in TAC1, G583R and S862C in the CpMRR1 etc. If lab routine workflow does include searching for these mutations associated with azole-resistance it should be acknowledged.

Answer: We thank the reviewer for this relevant comment. We did not perform molecular testing to assess specific mutations associated with azole resistance (such as ERG11 Y132F, TAC1 K143R/R398I/G650E, or CpMRR1 G583R/S862C) as this analysis is not part of our laboratory’s routine workflow. We have now acknowledged this as a limitation of our study and included a statement in the Discussion section of the revised manuscript.

  • Line 115-123. Were any genome sequencing analyses performed? If not please acknowledge.

Answer: We thank the reviewer for this pertinent comment. No genome sequencing analyses were performed.

Results.

  • Line 152. What about the other antifungal molecules tested? Were voriconazole and posaconazole affected too? Authors should report antifungal susceptibility profiles of all molecules tested.

Answer: We thank the reviewer for this comment. In our laboratory, antifungal susceptibility testing is routinely performed only for fluconazole; other antifungal agents such as voriconazole or posaconazole are not tested in our routine workflow.

  • Did fluconazole resistance increase over-time? Authors should depict antifungal resistance incidence rate over time in a separate figure, this is a very important aspect that in a study like this has to be fully elucidated.

Answer: We thank the reviewer for this valuable comment. We agree that temporal trends in fluconazole resistance are an important aspect to be addressed. In response, we have now analyzed the annual distribution of fluconazole-resistant C. parapsilosis isolates throughout the study period and included these data as a new figure (Figure 2C) in the Results section.

  • Figure 1. How many isolates were included in the dendrogram study? How did the authors select these isolates? If this is a retrospective study how was it possible to select such isolates? Authors must explain this important part of the study in both results and methods section.

Answer: We thank the reviewer for this important comment. A total of 17 C. parapsilosis isolates were selected for molecular analysis based on their availability and preservation in our laboratory strain collection, where representative isolates are routinely stored at −80 °C in glycerol stocks. We have now included this information in both the Methods section (130-132) and the Results section of the revised manuscript to clarify how the isolates were selected for molecular analysis despite the retrospective nature of the study.

  • Figure 3. Candida parapsilosis and Candida albicans are indistinguishable, please correct the figure accordingly.

Answer: We thank the reviewer for this observation. We have revised Figure 3 to ensure that C. parapsilosis and C. albicans are clearly distinguishable, using distinct colors and labels. The corrected version has been included in the revised manuscript (Figure 2B).

  • Table 1. this table is too big, either authors place it in a supplementary table or divide it into separate tables.

Answer:  We sincerely thank the reviewer for this valuable observation and for the overall positive evaluation of our manuscript. We agree that Table 1 was extensive and could be difficult to navigate. To improve readability and clarity, we have reorganized the table by grouping related variables into thematic sub-sections (demographic characteristics, comorbidities, clinical features, microbiological findings, treatment variables, and outcomes). These changes have been implemented in the revised version of the manuscript (now Table 1and 2).

Discussion.

  • Line 272. Is 81% a prevalence or a resistance rate? From my understanding this percentage is an overall calculation, however incidence rates have not been reported or assessed through the study. Authors are encouraged to better clarify resistance rates and trends overtime as well as clarifying and describing resistance rate for other molecules tested.

Answer: We thank the reviewer for this observation. The sentence was removed from the revised manuscript because it was imprecise in terminology (referring to prevalence instead of resistance rate) and redundant with information already presented in the first paragraph of the Results section. Additionally, as requested, we have now analyzed and presented the yearly distribution of fluconazole-resistant isolates to better illustrate temporal trends (Figure 2C). Regarding other antifungal agents, antifungal susceptibility testing is not routinely performed for molecules other than fluconazole in our laboratory; therefore, resistance rates for other antifungals could not be assessed. This has now been clarified and acknowledged as a limitation in the Discussion section.